# Drift Error Compensation Algorithm for Heterodyne Optical Seawater Refractive Index Monitoring of Unstable Signals

**DOI:** 10.3390/s23208460

**Published:** 2023-10-14

**Authors:** Shiwen Zhang, Liyan Li, Yuliang Liu, Yan Zhou

**Affiliations:** 1Optoelectronics System Laboratory, Institute of Semiconductors, Chinese Academy of Sciences, Beijing 100083, China; shiwenzhang@semi.ac.cn (S.Z.); ylliu@semi.ac.cn (Y.L.); zhouyan@semi.ac.cn (Y.Z.); 2College of Materials Science and Optoelectronics Engineering, University of Chinese Academy of Sciences, Beijing 100049, China

**Keywords:** drift error compensation, heterodyne interferometer, seawater refractive index, unstable signal

## Abstract

The refractive index measurement of seawater has proven significance in oceanography, while an optical heterodyne interferometer is an important, highly accurate, tool used for seawater refractive index measurement. However, for practical seawater refractive index measurement, the refractive index of seawater needs to be monitored for long periods of time, and the influence of drift error on the measurement results for these cases cannot be ignored. This paper proposes a drift error compensation algorithm based on wavelet decomposition, which can adaptively separate the background from the signal, and then calculate the frequency difference to compensate for the drift error. It is suitable for unstable signals, especially signals with large differences between the beginning and the end, which is common in actual seawater refractive index monitoring. The authors identify that the primary cause of drift error is the frequency instability of the acousto-optic frequency shifter (AOFS), and the actual frequency difference was measured through experimentation. The frequency difference was around 0.1 Hz. Simulation experiments were designed to verify the effectiveness of the algorithm, and the standard deviation of the optical length of the results was on the scale of 10^−8^ m. Liquid refractive index measurement experiments were carried out in a laboratory, and the measurement error was reduced from 36.942% to 0.592% after algorithm processing. Field experiments were carried out regarding seawater refractive index monitoring, and the algorithm-processing results are able to match the motion of the target vehicle. The experimental data were processed with different algorithms, and, according to the comparison of the results, the proposed algorithm performs better than other existing drift error elimination algorithms.

## 1. Introduction

The density and salinity of seawater are important environmental parameters in oceanography, and measuring them accurately is of great significance for underwater navigation, currents and underwater weather forecasting, marine ecological research, and seabed resource exploration. Due to the limitations of the measurement mechanism, a traditional Conductivity–Temperature–Depth (CTD) device cannot react to the non-ionic substances in seawater and, thus, cannot obtain information about the absolute salinity and real density of seawater. Moreover, this measurement method is very sensitive to temperature, and it is difficult to achieve a high measurement accuracy. The method of obtaining the density and salinity of seawater by measuring the refractive index of the seawater is gradually becoming a research hotspot in various oceanographic fields. Using the seawater refractive index measurement has many advantages, such as higher temperature stability [1] and closer relationships with the density and absolute salinity of seawater [2,3].

Among various measurement techniques, the optical heterodyne interferometer technique is one of the mainstream optical measurement techniques for a high-accuracy seawater refractive index measurement, due to its advantages of being contactless and less affected by impurities. Hiroshi Uchida et al. obtained seawater refractive index measurements with a standard deviation of $2.93\times 10^{-8}\;RIU$ using the optical measurement technique [4]. And novel interferometry structures can achieve even higher displacement accuracy, within the order of sub-nanometers [5], which makes it possible to develop an ultra-high-resolution seawater refractive index measurement device.

However, there are also some mechanisms that affect the accuracy of a seawater refractive index measurement system, including the uncertainty of laser frequencies, the nonlinearity of optical paths, noises (including shot noise, thermal noise, and electrical noise), uncertainty during signal processing, etc. Researchers have made efforts to eliminate those uncertainties during the development of high-accuracy measurement techniques [6]. 

As one of the influences of those uncertainties, a random drift error was found in the heterodyne interferometer measurement system in practical applications. Drift error can be very small, or even negligible in measurement applications over a short range or focusing on a high frequency. But, when it comes to long-term seawater refractive index monitoring, such as marine ecological environment and hydrological environment monitoring, the drift error of the measurement results will accumulate with time, which will cause a non-negligible distortion in the monitoring results. Independent studies of drift error, by B. K. A. Ngoi (1999) [7], L. Qian (2011) [8], and B. Lin (2022) [9], have shown that the instability of the optical shift of the AOFS is a primary cause of signal drift. 

Researchers have proposed various methods through which to remove this drift error; one method involves improving the optical path of the heterodyne interferometer system. B. J. Halkon et al. [10] added a corrective measurement path with which to calculate the actual real-time frequency shift of the AOFS and used the actual frequency shift to obtain the actual light length. However, the corrected measurement obtained through the corrected optical path needs to guarantee the covariance with the actual measurement optical path and focus on the static reference surface, which is almost impossible in its actual application. X. Zhang et al. designed two interferometric optical paths using optical fiber couplers and eliminated the drift error caused by frequency fluctuation through subtraction [11], but did not take the different clock systems of the two systems into account. These methods, through redesigned optical paths, have the same drift-error-removing effect for different types of signals. On the other hand, they are costly, complicated in structure, and introduce new sources of error.

Another method for eliminating drift error is using an algorithm. J. Shang et al. [12] obtained the actual frequency shift by extracting the drive current signal of the AOFS and obtained a measurement result based on the actual frequency shift. But, because of the delay between the interferometric signal and the drive signal, the practical effect of the algorithm was limited. The ratio of the displacement error to the real value they finally obtained was less than 4.6%. Q. Zheng et al. designed a dynamic Fourier filter with which to eliminate the ambient noise, including drift error [13], but this method is only applicable to narrow-band sinusoidal signals, which is restrictive in practical applications. A. Darwish and other researchers preprocessed their signals by subtracting the first-order least squares fitting value, i.e., linear fitting compensation algorithms, to remove the drift error [14]. In addition, other researchers used wavelet filtering to remove low-frequency noise [15,16], which also achieves the same effect.

Drift error removal algorithms based on linear fitting compensation, Fourier filtering, and wavelet filtering are mainly applicable to normal signal processing with strong stability, such as a low-fluctuation signal with an equilibrium position. But, in the case of an unstable signal with strong randomness, especially when there is a large difference between the starting point and the ending point of the signal, it is not possible to achieve drift error elimination using the currently available algorithms. In seawater refractive index monitoring, the measurement result can be considered an unstable signal with strong randomness. There is no reliable equilibrium position in monitoring the signal, and large fluctuations occur from time to time. As a result, the existing signal drift elimination algorithms cannot meet the requirements of seawater refractive index monitoring.

To solve the problems of heterodyne interferometers in seawater refractive index monitoring, the authors of this paper propose a drift error compensation algorithm based on wavelet decomposition. To meet the application of long-term monitoring, the authors use the Otsu method for threshold selection, thus adaptively distinguishing between the signal region and the non-signal region through wavelet analysis. By analyzing the non-signal region, the difference between the actual frequency shift and the set frequency shift of the AOFS (called the frequency difference) is obtained, which, in turn, compensates for drift errors. Compared with the existing linear fitting compensation techniques, Fourier filtering and wavelet filtering algorithms, the algorithm proposed is able to effectively remove the drift error from unstable signals and has strong adaptability and robustness.

In addition, the authors of this paper analyze the magnitude and characteristics of the drift error introduced by using the AOFS, according to the actual heterodyne interferometer system. According to the real monitoring data, the slicing signal length (3 min) and the criterion of successful classification (the background region being no less than 1/10 of the signal length) are set for the algorithm, and simulation experiments are designed to verify the effectiveness and adaptability of the algorithm. In the experiments on liquid refractive index monitoring in the laboratory and on seawater refractive index monitoring in the external field, good results and algorithm comparisons verify the strong practicality of the proposed algorithm.

## 2. Principle of Optical Seawater Refractive Index Measurement

The optical seawater refractive index measurement adopts the laser heterodyne interferometer technique, and its schematic diagram is shown in Figure 1.

The light emitted from the laser is divided into a measurement beam ($P_{m0}$) and a reference beam ($P_{r}$) via the polarizing beam splitter 1 (PBS 1). The measurement beam is projected onto a fixed target through a seawater measurement interval of length *D*, and reflected back to polarizing beam splitter 2 (PBS2), during which the power of the measurement beam decays to $P_{m}$. The reference beam goes through the AOFS, where its frequency is shifted to a higher level. Because the measurement beam passes through the quarter-wave plate twice, its polarization angle is the same as that of the reference beam, which ensures the generation of interference light at the beam splitter (BS). The optical lens (L) is used for beam focusing. The photodetector (Detector) receives the interference light and converts it into an electrical signal ($i_{raw}$). The entire optical design is based on Mach–Zehnder interferometry, which is widely used for refractive index measurement. Compared with the traditional structure, the existence of the AOFS makes it possible to collect data in a wide band and to distinguish the direction of the phase change.

Due to the frequency instability of the acousto-optic frequency shifter, the actual frequency shift is not equal to the preset frequency shift; the actual shift of the acousto-optic frequency was $f_{A}$, and the preset frequency shift was $f_{0}$.

The raw interference photocurrent $i_{raw}$ at the photodetector can be expressed as:(1)$$i_{raw}\left(t\right)=\eta P\left(t\right)=\eta [P_{m}+P_{r}+2\sqrt{P_{m}P_{r}}\cos\left(2\pi f_{A}t-\frac{4\pi l(t)}{\lambda }+\varphi_{0}\right)]\;$$
where η is the photoelectric conversion efficiency, l(t) is the optical length of the measurement interval, and $\varphi_{0}$ is the initial phase.

The obtained raw photocurrent signal $i_{raw}\left(t\right)$ is demodulated via a Differentiate-and-Cross-Multiply approach, or an arctangent approach, to obtain the phase change value ∆ϕ, as follows:(2)$$∆\phi (t)=2\pi \left(f_{0}-f_{A}\right)t+\frac{4\pi (l\left(t\right)-l\left(0\right))}{\lambda }$$

Ideally, $f_{A}$ equals $f_{L}$, and the variation measurement of optical length ∆l′ is obtained as follows:(3)$$∆l\prime =\frac{\lambda \cdot ∆\phi (t)}{4\pi }$$

When a frequency difference exists, the actual variation in optical length ∆l should be $∆l^{\prime }$, subtracted from the influence of the frequency difference in the AOFS:(4)$$∆l=∆l^{\prime }-\frac{\lambda }{2}\left(f_{0}-f_{A}\right)t=\frac{\lambda \cdot ∆\phi (t)}{4\pi }-\frac{\lambda }{2}\left(f_{0}-f_{A}\right)t$$

According to the definition of a refractive index, the variation in the seawater refractive index in the measurement interval ∆n is proportional to the actual variation in optical length ∆l; if we let the physical length of the measurement interval be *D*, then ∆n can be expressed as
(5)$$∆n=\frac{∆l}{D}$$

From Formula (4), it is obvious that the drift error introduced due to the frequency instability of the AOFS is $-\frac{\lambda }{2}\left(f_{0}-f_{A}\right)t$. The value of the drift error directly relates to the frequency difference ($∆f=f_{0}-f_{A}$). In the actual experiment, the frequency difference ∆f can be obtained by recording a signal for a period of time (∆t) in a calm environment; ensuring that there is no optical length change during that time period, ∆l=0: (6)$$∆l^{\prime }=∆l+\frac{\lambda }{2}\left(f_{0}-f_{A}\right)∆t=0+\frac{\lambda }{2}\left(f_{0}-f_{A}\right)∆t$$

Thus, the frequency difference is
(7)$$∆f=\frac{2}{\lambda }\cdot \frac{∆l^{\prime }}{∆t}$$

Once the frequency difference is obtained, it is possible to compensate for the drift errors and obtain the actual variation in the optical length. Thus, the variation in the seawater refractive index can be calculated according to Formula (5).

It is worth mentioning that fluctuations in laser frequency can cause measurement error, but that error would not accumulate. According to Equation (2), at every moment when the wavelength of the laser is equal to the nominal wavelength, the error caused by fluctuating laser frequency vanishes. In addition, the power of the light is not stable either, but the Automatic Gain Control (AGC) circuit behind the detector will somehow eliminate its influence.

## 3. Principle of Discrete Wavelet Decomposition

Wavelet analysis has a unique advantage for the time–frequency analysis of signals because of its more adaptive time domain resolution and frequency domain resolution compared to those of the Fourier transform technique. For orthogonally normalized wavelet bases, in addition to the mother wavelet, a father wavelet, which is also called the scaling function, also exists, and it is the key concept of multiresolution analysis (MRA).

The mathematical concept Lebesgue space, $L^{p}(R)$, refers to the space of functions consisting of p times integrable functions. In the field of signal processing, a signal can be considered as a function of time. In this paper, we assume that the obtained signal f(t) satisfies $t\in R,f(t)\in L^{2}(R)$, i.e., the signal f(t) is a measurable function that is square-integrable everywhere for R, which physically represents the restriction of the signal; the signal energy at every moment must be finite. Signals of infinite energy, such as the Impulse Function, are not square-integrable.

According to the theory of multiresolution analysis, there are many approximate subspaces in the $L^{2}(R)$ space satisfying
(8)$$\ldots \subset V_{-1}\subset V_{0}\subset V_{1}\ldots \subset L^{2}(R)$$
(9)$$f\left(t\right)\in V_{j}⟺f\left(2t\right)\in V_{j+1},\;\forall j$$

The approximate subspace $V_{j}$ is actually a coarse representation of $L^{2}(R)$, and larger values of *j* indicate more refined functions in $V_{j}$. In practical signal processing, time cannot traverse every real number, but is usually discrete instead, so the practical signal is often an element of an approximate subspace with a certain fineness *J*, i.e., $f\left(t\right)\in V_{J},\;\exists J$.

The orthonormal basis of the approximate subspace consists of the scale functions, and with the set of scale functions $V_{J}$, the original signal $f\left(t\right)$ can be expressed in the linear combination of $\varphi_{i.j}\left(t\right)\in V_{J}$. But, if we want to describe the more refined signal $f\left(2t\right)\in V_{J+1}$, we need to add new functions to the set of scale functions of $V_{J}$ to form the orthonormal basis of the approximate subspace $V_{J+1}$. Omitting the complicated mathematical proof, the wavelet functions $\varphi_{k}\left(t\right)$ of the $V_{J}$ layer are the required new functions. Let the space spanned with the wavelet functions as orthonormal basis be $W_{J}$, and the relationship between $V_{J}$ and $V_{J+1}$ satisfies the following: (10)$$V_{J+1}=V_{J}⨁W_{J}$$

Further,
(11)$$V_{J}=V_{0}⨁W_{0}⨁W_{1}⨁\ldots ⨁W_{J-1}$$

In the MRA method, the approximate subspace $V_{J}$, in which the original signal is located, is decomposed as follows, according to Equation (10):(12)$$V_{J}=V_{J0}⨁W_{J0}⨁W_{J0+1}⨁\ldots ⨁W_{J-1}$$

The right-hand side of Equation (12) contains the coarsest approximate subspace $V_{J0}$, and a series of spaces spanning the wavelet functions. Discrete wavelet decomposition calculates the convolution of the signal and the wavelet functions of each space layer to obtain the wavelet decomposition coefficients at each layer and time. The coefficients, to some extent, represent the local oscillation intensity of the signal.

## 4. Algorithm Design

In the application of seawater refractive index monitoring, the recorded signal could be regarded as a background signal in a calm environment most of the time, which means that the optical length variation is 0, according to Formula (6). When the refractive index of seawater in the measurement interval changes, the signal fluctuates more fiercely than the background signal does. The discrete wavelet decomposition of the recorded signal when the signal changes obviously has larger decomposition coefficients compared with those in the background signal wavelet decomposition. According to this law, in this paper, the authors considered the time period with larger coefficients as the fluctuation region, and the time period with smaller coefficients as the background region, and assumed the variation in optical length in the background region to be 0, ∆l=0. The frequency difference was calculated from the signal in the background region, and then the drift error of the signal was compensated for.

For implementation, the algorithm has a Daubechies wavelet basis, which is a standard orthogonalized wavelet basis commonly used for signal decomposition and reconstruction. For each layer of discrete wavelet decomposition coefficients, a threshold needs to be selected in order to judge whether the coefficients are large or not. Since the purpose of the threshold is to classify signals, the maximum between-cluster variance method (Otsu method) [17] was chosen to verify the threshold value. In this method, coefficients larger than the threshold value correspond to the fluctuation region, and coefficients smaller than the threshold value correspond to the background region. In addition, for each time period, it is necessary to determine whether there is a distinction between the fluctuation region and the background regions, and for cases where there is no distinction, the frequency difference from the previous time period is recommended. The block diagram of the algorithm procedure is shown in Figure 2.

## 5. Simulation

### 5.1. Simulation Conditions

In this paper, we used a laser heterodyne interferometer device, as shown in Figure 3. The light source was a He-Ne laser with standard wavelength 632.8 nm, output power > 10 mW, linewidth < 0.1 kHz and optical signal noise ratio > 50 dB. The characteristics of the photodetector are shown in Table 1.

The AOFS used in the laser heterodyne interferometer system has an inbuilt low-power signal generator, which produces an RF signal at a fixed frequency. The range of the generated signal frequency indicates the stability of the generator. The signal generator used in the experimental device is Gooch & Housego’s AODR 97-03307-74 (1040AF-AIF0-0.5) [18], with a center frequency of 40 MHz ± 0.1%, and according to the parameters of the frequency error range of the signal generator, the actual frequency difference $∆f_{practical}$ satisfies
(13)$$-0.1\%\times 40\;MHz=-40000\;Hz<∆f_{practical}<40000\;Hz=+0.1\%\times 40\;MHz$$

Fifty signal acquisition tests in calm environments were executed at different times and locations, and the frequency difference from each test was obtained according to Equation (7), as shown in Appendix A. The average value of all of the frequency differences was $-1.896\times 10^{-8}\;Hz$*,* which is negligible. The maximum value of the frequency differences was 0.1725 Hz, and the minimum value was −0.1093 Hz. All of the frequency differences were within the limits of frequency error shown in Equation (13). This means that, without considering issues such as equipment aging or temperature effects, a qualified AOFS in a laser heterodyne interferometer system is also likely to produce the drift error shown in this paper.

The refractive index measurement experiment, using a distilled water sample, was carried out for a long period of time (2 h and 26 min) to obtain the variation in frequency difference over time, and the result is shown in Figure 4.

According to the experimental results, it can be seen that the overall variation range of the frequency difference is in the order of 0.1 Hz, and the frequency difference changes slowly over time. In order to achieve good drift error compensation results, the slicing signal length was set to be 3 min in the algorithm.

Although the Otsu method is an effective threshold selection strategy for discrete wavelet decomposition coefficients, it cannot judge whether a signal satisfies the classification condition, i.e., the signal contains both background signals and fluctuating signals. Based on several experiments, the artificially set length of the background time region needs to be at least 1/10 of the slice time; otherwise, the background region will be too scattered, and the stored frequency difference will be used for compensation processing in the algorithm. This setting prevents the algorithm from compensating useful information when faced with long fluctuating signals.

### 5.2. Simulation Experiment

When the drift error introduced by a frequency difference of 0.1 Hz was simulated, the wavelength was set to 632.8 nm, and the signal length was set to 3 min, in order to have better agreement with the parameters of the device used.

In order to simulate actual measurement signals, white noise with a standard deviation of $10^{-8}\;m$*,* which is a common white noise magnitude in laser heterodyne interferometer measurements, was added. 

To identify various signals, the simulation experiments mainly compared the drift error elimination performances of the linear fitting compensation algorithm, the Fourier filtering algorithm (high-pass filtering with a cutoff frequency of 0.5 Hz), the wavelet filtering algorithm (wavelet high-pass filtering with db4 wavelet basis and depth of 12 layers), and the method proposed in this paper.

The performances of various algorithms were quantified using the standard deviation σ between the algorithm-processed signal and the ideal signal, as follows: (14)$$\sigma =\sqrt{\frac{\Sigma {\left(y_{i}-\hat{Y_{i}}\right)}^{2}}{n}}$$
where $y_{i}$ is the ideal signal value and $\hat{Y_{i}}$ is the processed signal value.

In this simulation experiment, the ideal signal represents an actual variation in optical length, as shown in Equation (4). As verified in Section 2, the variation in the refractive index, which is the purpose of measurement, is equal to the actual variation in the optical length divided by measurement interval length *D*. So, the variation in the ideal signal is in direct proportion to the variation in the refractive index. The standard deviation σ is the quantization of the closeness of the algorithm process result with the actual variation in the optical length. The smaller the standard deviation is, the better the algorithm performs. For other applications, the optical length might indicate other physical quantities.

#### 5.2.1. Comparison of Drift Error Elimination Performances in a Non-Disturbed Environment

In this simulation experiment, the performances of various algorithms in a calm environment were analyzed without adding any other signals except the drift signal, due to the addition of the simulated frequency difference of 0.1 Hz and the white noise signal. This condition corresponded to the refractive index measurement of a liquid sample. The simulation results are shown in Figure 5.

The standard deviations σ between the processed results of the various methods and the ideal signal are shown in Table 2.

As can be seen from Table 1, all four algorithms work well in an unperturbed state, and the standard deviation between the obtained final algorithmic processing results and the ideal signal (constant 0 signal) is basically of the same magnitude as that with added white noise. In other words, for liquid samples, all tested algorithms perform well. 

#### 5.2.2. Comparison of Drift Error Elimination Performances for a Sinusoidal Fluctuating Signal with the Same Start and End Values

A sinusoidal signal was used as an example with which to study the performances of various algorithms when the ideal signal is periodically oscillating, which is the characteristic of the measurement signal in many application scenarios, so the drift error removal performances in this case are very important. It is worth mentioning that the ideal signal in this simulation experiment starts at 0 and finally returns to 0, i.e., the start point and the end point have the same value. In addition to the 0.1 Hz drift signal and the white noise signal, a sinusoidal signal with a frequency of 1 Hz and an amplitude of 5 μm was added, with a length of 60 periods, i.e., 60 s. The amplitude was set to meet the amplitude of many mechanical vibrations, which is in the order of micrometers. While this frequency is relatively low, for a wide range of high-pass filtering algorithms, low-frequency cases are always more difficult to deal with. For refractive index measurement, it is hard to imagine what this condition corresponds to, as the ideal signal often appears in displacement or velocity measurements. The simulation results are shown in Figure 6.

The standard deviations σ between the processed result of the various methods and the ideal signal are shown in Table 3.

As can be seen from Table 2, the standard deviations of the results of the Fourier filtering algorithm and wavelet filtering algorithm in this case are much larger than those of the linear fitting and proposed methods, but are still in the order of $10^{-8}\;m$. This indicates that the algorithm processing results are still in good conformity with the ideal signal. As can be seen in Figure 6, there is a small discrepancy between the processed results and the ideal signal when the signal suddenly changes.

#### 5.2.3. Comparison of Drift Error Elimination Performances for a Sinusoidal Fluctuating Signal with Different Start and End Values

In order to study the performances of various algorithms in the case that the ideal signal is a periodic oscillating signal, but does not end up at the equilibrium value, the length of the ideal signal is 59.75 periods, with the end value being −5 μm. Similarly, the ideal signal is a sinusoidal signal with a 1 Hz frequency and a 5 μm amplitude. When monitoring vibration signals, the vibration source often does not return to the equilibrium position when it stops, and this signal characteristic is often ignored by researchers. In seawater refractive index monitoring applications, for irregular signals, it is almost impossible that the start signal value and the end value are equal, so the algorithm performance needs to be evaluated for cases in which there is a large difference between the start and end points of the signal. Similarly, for refractive index measurement, it is hard to imagine what this condition corresponds to. The simulation results are shown in Figure 7.

The standard deviations σ between the processed results of the various methods and the ideal signal are shown in Table 4.

From Figure 7, it can be seen that the linear fitting compensation algorithm is unable to obtain an efficient fitting line, while the Fourier filtering and wavelet filtering algorithms completely ignore the signal feature, so that it stays at −5 μm after 120 s. It can also be seen from Table 3 that the three algorithms, but not the proposed method, have standard deviation values in the order of $10^{-6}\;m$, which is in the same order as the signal. This result indicates that current algorithms are poorly adapted to practical applications. In contrast, the algorithm proposed in this paper is able to adaptively extract the frequency difference in the system and finally obtains a standard deviation with a $10^{-8\;}\;m$ magnitude. That is, even for displacement or velocity measurements, the existing algorithms cannot meet the requirements for drift error elimination.

#### 5.2.4. Comparison of Drift Error Elimination Performances for a Signal Containing Sudden Changes

In order to study the performances of various algorithms with the existence of sudden signal changes, a positive-direction signal change of 5 μm was added at 60 s, and a negative-direction signal change of 5 μm was added at 120 s into the ideal signal. In actual seawater refractive index monitoring, the pressure, temperature, salinity, and other external factor changes will have an impact on the seawater refractive index. The optical length will suddenly change in one direction with the appearance of external factors, and it will change back with the disappearance of the external factors, which is in line with the setup of this simulation. For refractive index measurement, this condition corresponds to measuring an internal solitary wave, which is important in oceanography. The simulation results are shown in Figure 8.

The standard deviations σ between the processed results of the various methods and the ideal signal are shown in Table 5.

It can be seen from Figure 8 that the linear fitting compensation algorithm shifts the overall signal in the negative direction on the y axis, since, from 60 s to 120 s, the signal remains positive. The results of the Fourier and wavelet filtering algorithms not only fluctuate near the sudden signal change, but also fail to restore the ideal signal value +5 μm at 60–120 s. It can also be seen from Table 4 that the method proposed in this paper demonstrates a lower standard deviation than those of the existing methods in this simulation condition. Thus, a conclusion can be drawn that, to measure the refractive index of a solitary wave in the ocean, the proposed adaptive compensation algorithm is the most effective choice.

## 6. Experimental Verification

### 6.1. Laboratory Liquid Refractive Index Measurements

The experimental setup is shown in Figure 9. The glass tank is 22.3 cm long and 15.2 cm wide, i.e., *D* = 15.2 cm in Equation (5). The distilled water was put into the glass tank at a height of 10.0 cm, there was a total of 3389.6 mL of water in the tank, and the temperature of the water was 15.6 °C. A volume of 33.9 mL of standard sample seawater, with a calibrated salinity of 35‰, was removed using a measuring cylinder and a rubber-tipped burette. Signal acquisition was performed first, and the seawater sample was poured into the glass tank after 60 s of the acquisition process, which ended at 180 s. The experimental data are shown in Figure 10.

According to the temperature–salinity–refractive index table and the fitting formula of these three parameters, the refractive index of pure water at 15.6 °C is 1.331201 RIU at atmospheric pressure, and the refractive index of salt water with a salinity of 0.35‰ at 15.6 °C is 1.331264 RIU. According to Formula (5), substituting *D* = 15.2 cm, the optical length variation obtained with the heterodyne interferometer measurement system should be $9.58\times 10^{-6}m$. Before drift error compensation, the optical length variation at 180 s was $1.3119\times 10^{-5}m$, and after being processed with the proposed algorithm, the optical length variation changed to $9.5233\times 10^{-6}m$. The measurement error was reduced from 36.942% to 0.592%. This shows that the proposed drift error compensation algorithm is effective in the application of liquid refractive index monitoring.

On the other hand, the raw experimental data were processed with other drift error elimination algorithms as well. The results are shown in Figure 11.

According to Figure 11, one can tell that the existing algorithms could not deal with an unstable signal well. For the linear fitting compensation algorithm, the processed result came out with a negative value at the beginning, which is not reasonable. For the Fourier filtering and wavelet filtering algorithms, the filters caused too much information loss, which departs from the purpose of the measurement. While the result of proposed adaptive compensation algorithm turned out to be in conformity with the theoretical calculation, the three existing algorithms turned out to be noneffective with their different results.

### 6.2. Field Experiment of Seawater Refractive Index Measurements

The field experiments on seawater refractive index monitoring were conducted in Tianjin, China, and the experimental scenario is shown in Figure 12. The device was fixed behind the vessel using a rope, and the traveling state of the vessel and the acquisition time were recorded to be analyzed later. A background signal of 5 min appeared at the beginning of the process, and then the vessel traveled with the device in tow for 10 min. After this, the vessel stopped, and signal acquisition continued for 15 more minutes. The vessel moved twice to different positions. During experiment 1, the vessel moved away from the shore; for experiment 2, the vessel was far from the shore the entire time. The experimental raw data and the results processed with the algorithm are shown in Figure 13.

From Figure 13, it can be seen that the data between the 300 s and 900 s range are very different from the background signals, a result which is in good agreement with the traveling state records, indicating that the refractive index monitoring device can detect tiny seawater refractive index changes caused by traveling motions. Moreover, this indicates that the drift error compensation algorithm proposed in this paper can effectively remove the drift error generated during the acquisition process, which improves signal reliability in the application of seawater refractive index monitoring.

Similarly, the effects of existing algorithms were investigated under the same field experiment conditions. The results are shown in Figure 14.

It is important to mention that, in this part, there are no criteria for judging which result is the best result. To distinguish the movement condition of the vessel, all of these algorithms can somehow achieve the intended goal. From Figure 14, when the vessel is towing the device, the fluctuation in the signal is strong; this characteristic remains after being processed with every tested algorithm. However, the information on variation is lost after processing with the Fourier filter or wavelet filter algorithms. Only the linear fitting compensation algorithm retains the information on variation. In Figure 14b, the result of the linear fitting compensation algorithm is similar to the result of the proposed adaptive algorithm. The reason for this is that the variation trend of this particular signal is consistent, which reveals the fact that the linear fitting compensation algorithm is designed to deal with stable signals.

## 7. Conclusions

The authors of this paper analyzed the influence of AOFS frequency instability on drift error and evaluated the frequency difference variation range of the actual AOFS device.

An adaptive drift error compensation algorithm, which divides the background and fluctuation regions of the signal through discrete wavelet decomposition and the Otsu method, and then compensates for drift error due to the instability of the AOFS, was proposed. Compared with the currently available drift error elimination algorithms, this algorithm performs better with unstable signals. The simulation experiments demonstrate that the standard deviation of the optical length between the signal processed with the proposed algorithm and the ideal signal reduces from the order of $10^{-6}\;m$ to the order of $10^{-8}\;m$ compared with the results of the existing algorithms. The proposed algorithm is effective for irregular signals and signals with sudden changes, which are common in seawater refractive index monitoring. The measurement error of the laboratory refractive index measurement was reduced from 36.942% to 0.592% after processing. In the field experiment, the algorithm increased the reliability of the seawater refractive index signal.

According to various comparisons, the existing drift error elimination algorithms fail to obtaining satisfactory results when the signal is unstable, but the proposed method is proven to be reliable. Furthermore, the proposed compensation algorithm does not require additional hardware and can also be used in a variety of laser heterodyne interferometer applications besides seawater refractive index monitoring, such as in the long-term micro-displacement monitoring of structures such as bridges, railways, and dams.

## Figures and Tables

**Figure 1 sensors-23-08460-f001:**
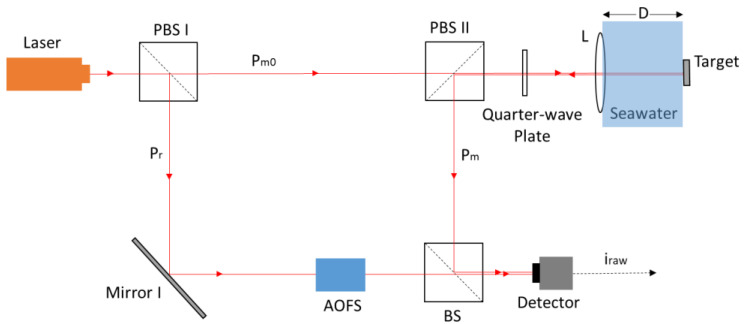
Schematic diagram of seawater refractive index measurement based on the laser heterodyne interferometer technique.

**Figure 2 sensors-23-08460-f002:**
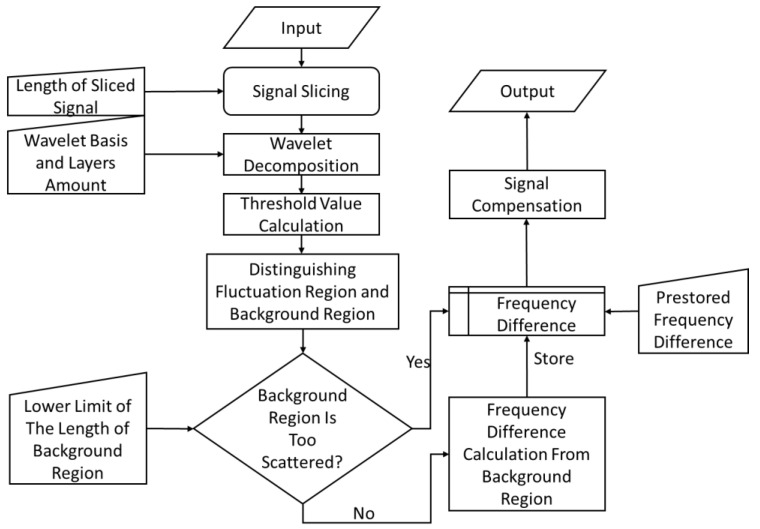
Block diagram of the drift error compensation algorithm procedure.

**Figure 3 sensors-23-08460-f003:**
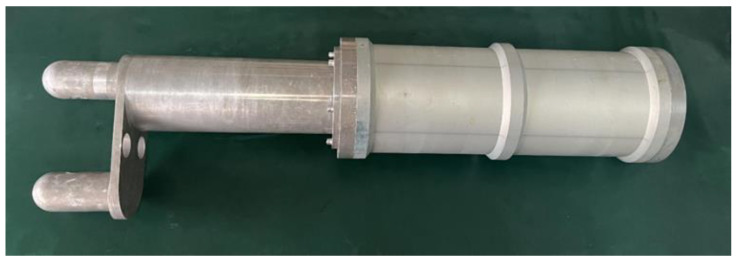
Photo of the experimental device.

**Figure 4 sensors-23-08460-f004:**
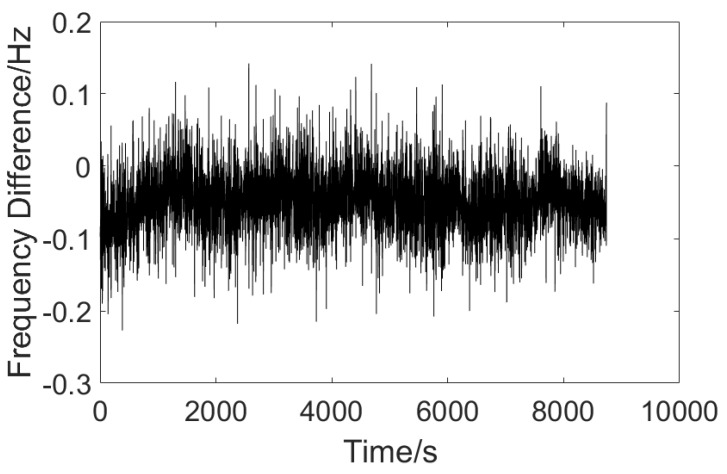
Plot of frequency difference over time.

**Figure 5 sensors-23-08460-f005:**
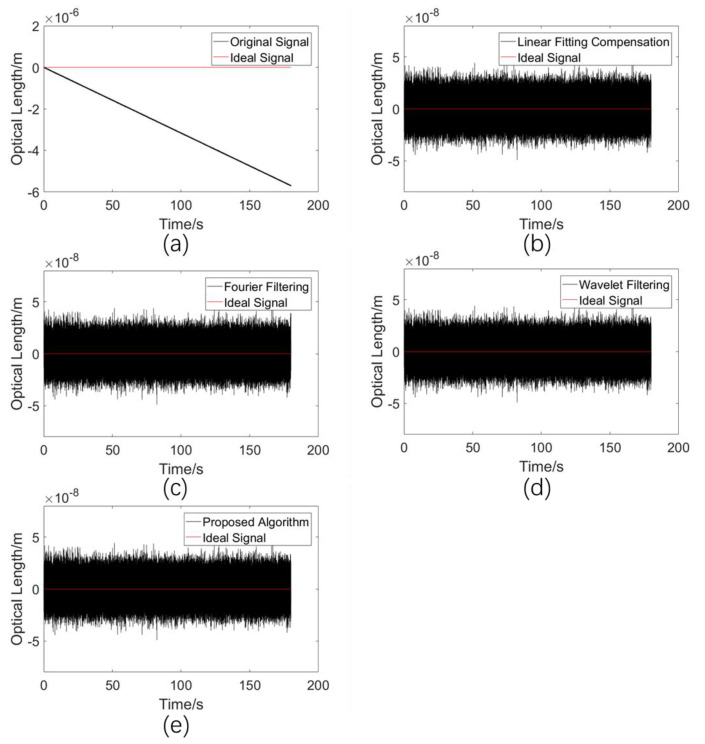
(**a**) Simulated drift error signal. (**b**) Processed results using the linear fitting compensation algorithm. (**c**) Processed results using the Fourier filtering algorithm. (**d**) Processed results using the wavelet filtering algorithm. (**e**) Processed results using the proposed adaptive compensation algorithm.

**Figure 6 sensors-23-08460-f006:**
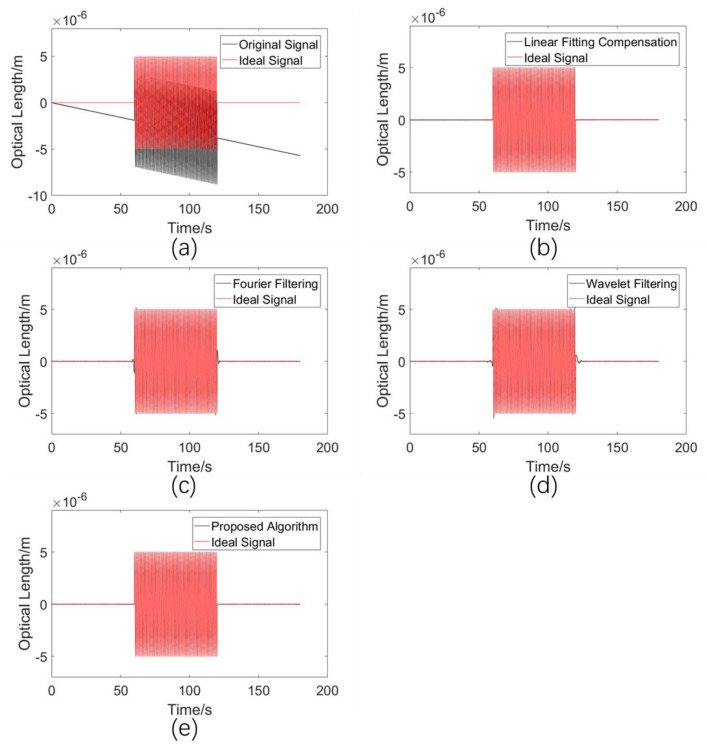
(**a**) Simulated drift error signal. (**b**) Processed results using the linear fitting compensation algorithm. (**c**) Processed results using the Fourier filtering algorithm. (**d**) Processed results using the wavelet filtering algorithm. (**e**) Processed results using the proposed adaptive compensation algorithm.

**Figure 7 sensors-23-08460-f007:**
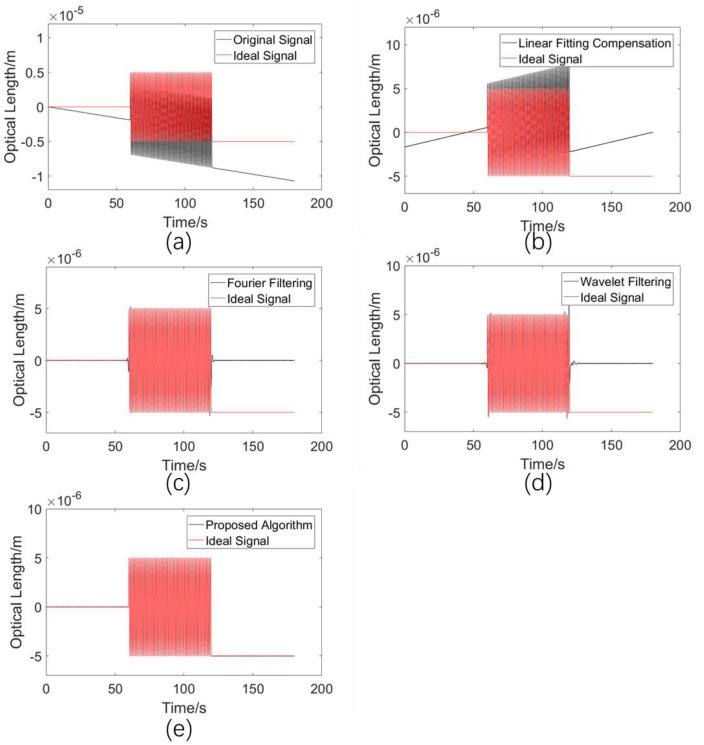
(**a**) Simulated drift error signal. (**b**) Processed results using the linear fitting compensation algorithm. (**c**) Processed results using the Fourier filtering algorithm. (**d**) Processed results using the wavelet filtering algorithm. (**e**) Processed results using the proposed adaptive compensation algorithm.

**Figure 8 sensors-23-08460-f008:**
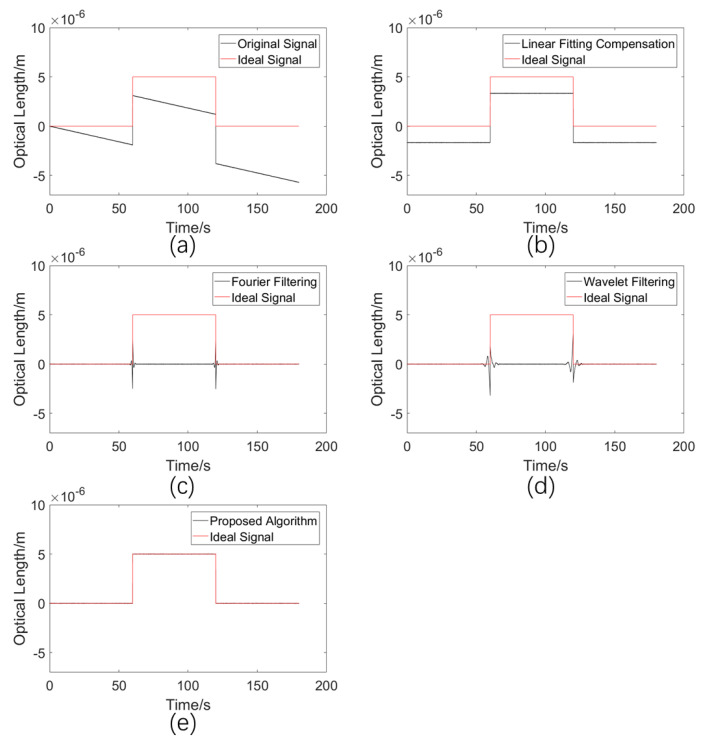
(**a**) Simulated drift error signal. (**b**) Processed results using the linear fitting compensation algorithm. (**c**) Processed results using the Fourier filtering algorithm. (**d**) Processed results using the wavelet filtering algorithm. (**e**) Processed results using the proposed adaptive compensation algorithm.

**Figure 9 sensors-23-08460-f009:**
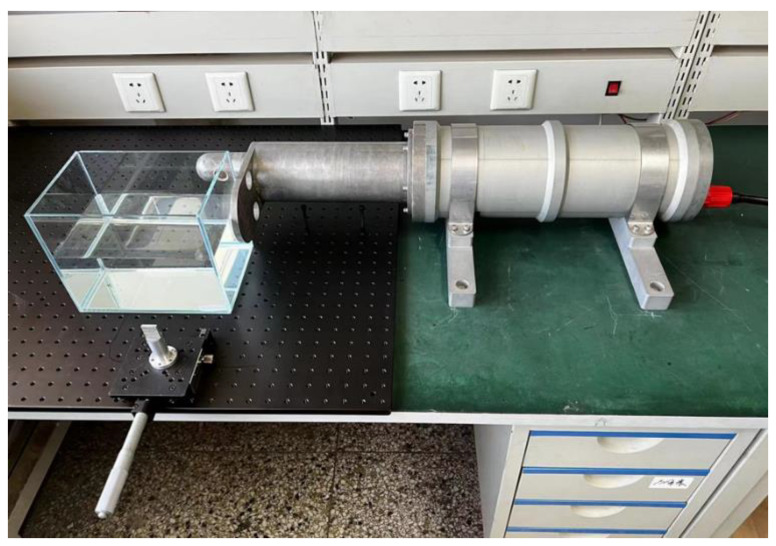
Experimental setup for liquid refractive index measurement.

**Figure 10 sensors-23-08460-f010:**
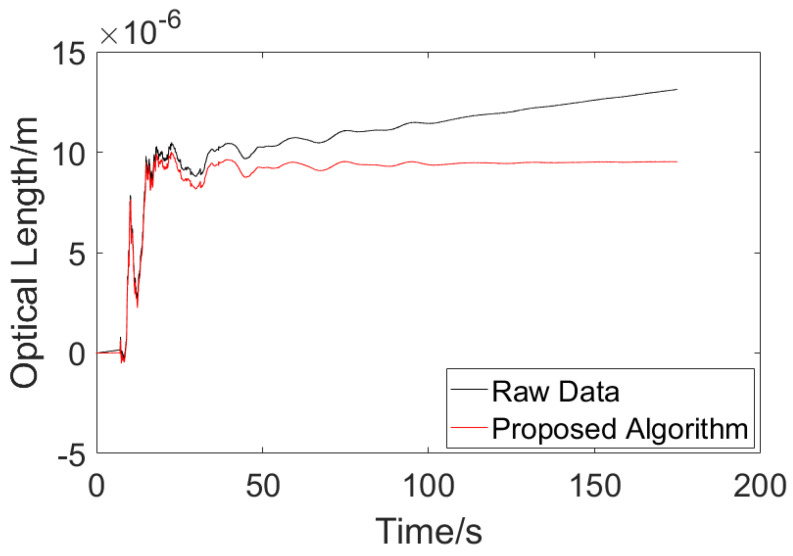
Refractive index measurement data.

**Figure 11 sensors-23-08460-f011:**
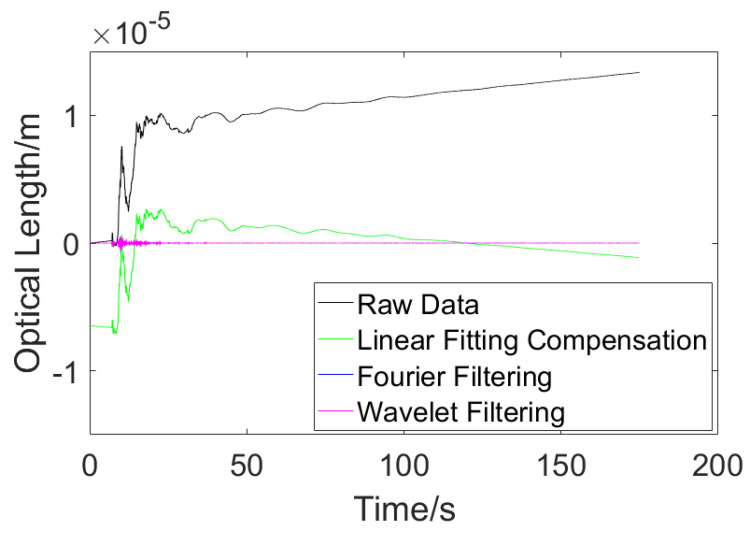
Results of existing drift error elimination algorithms.

**Figure 12 sensors-23-08460-f012:**
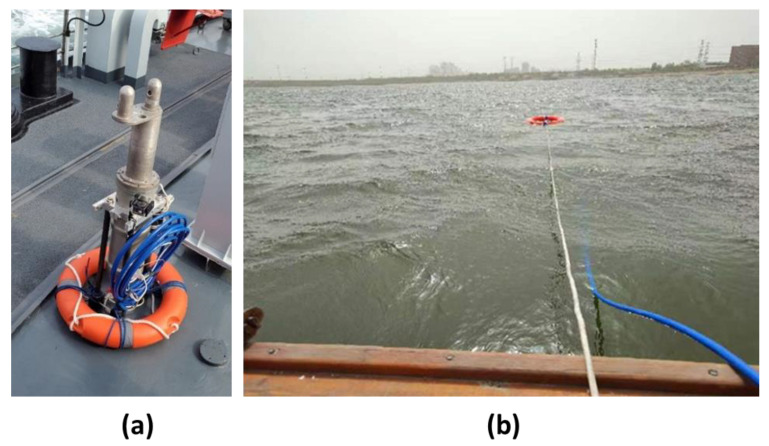
(**a**) Photo of the device. (**b**) Field experimental scenario.

**Figure 13 sensors-23-08460-f013:**
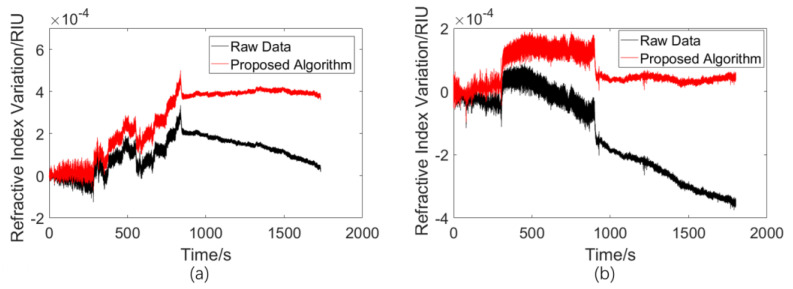
(**a**) Experimental result of moving away from the shore. (**b**) Experimental result of traveling far from the shore.

**Figure 14 sensors-23-08460-f014:**
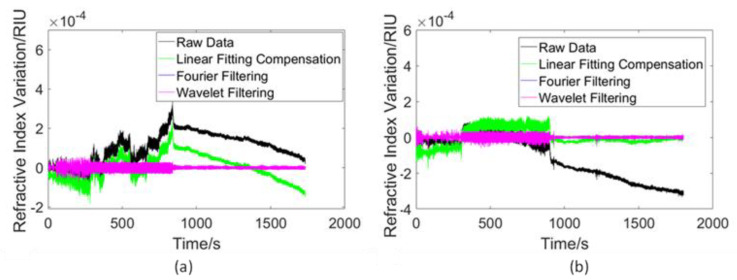
(**a**) Comparison of the algorithms’ performances on the data when moving away from the shore. (**b**) Comparison of algorithms’ performances on the data when traveling far from the shore.

**Table 1 sensors-23-08460-t001:** The characteristics of the photodetector.

Detector Model	Bandwidth (GHz)	Spectral Range (nm)	Quantum Efficiency@Peak	$\mathrm{Noise}\;\mathrm{Equiv}.\;\mathrm{Power}\;(W/\sqrt{Hz}$)
UPD-30-VSG-P@Alphalas	>10	320–900	40%	$3.0\times 10^{-15}$

**Table 2 sensors-23-08460-t002:** Standard deviations after algorithm processing.

Drift Error Elimination Algorithm	*σ* (m)
Linear fitting compensation	$1.0017\times 10^{-8}$
Fourier filtering	$1.0028\times 10^{-8}$
Wavelet filtering	$1.0023\times 10^{-8}$
Proposed adaptive compensation	$1.0017\times 10^{-8}$

**Table 3 sensors-23-08460-t003:** Standard deviations after algorithm processing.

Drift Error Elimination Algorithm	*σ* (m)
Linear fitting compensation	$1.1220\times 10^{-8}$
Fourier filtering	$9.7497\times 10^{-8}$
Wavelet filtering	$7.1236\times 10^{-8}$
Proposed adaptive compensation	$1.0017\times 10^{-8}$

**Table 4 sensors-23-08460-t004:** Standard deviations after algorithm processing.

Drift Error Elimination Algorithm	*σ* (m)
Linear fitting compensation	$2.5524\times 10^{-6}$
Fourier filtering	$2.8926\times 10^{-6}$
Wavelet filtering	$2.8906\times 10^{-6}$
Proposed adaptive compensation	$1.0019\times 10^{-8}$

**Table 5 sensors-23-08460-t005:** Standard deviations after algorithm processing.

Drift Error Elimination Algorithm	*σ* (m)
Linear fitting compensation	$1.6666\times 10^{-6}$
Fourier filtering	$2.8818\times 10^{-6}$
Wavelet filtering	$2.8766\times 10^{-6}$
Proposed adaptive compensation	$1.0018\times 10^{-8}$

## Data Availability

Not applicable.

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
