# Peer review of "Drift Error Compensation Algorithm for Heterodyne Optical Seawater Refractive Index Monitoring of Unstable Signals"

_sensors, 2023, doi:10.3390/s23208460_

Round 1

Reviewer 1 Report

The overall purpose of this paper is good and the logic is clear, the simulation method and scheme adopted by the author are feasible. The corresponding test has been verified, which has certain guiding significance for practical engineering application. The modification suggestions are as follows:

1. In the laboratory liquid refractive index measurement and seawater refractive index measurement, only the proposed drift error compensation algorithm is provided to suppress drift errors during the acquisition process. The effect is not compared with other drift error elimination algorithms of the same type. Therefore, the actual application effect and pros and disadvantages of the algorithm are not enough to prove.

2. In the process of laboratory and field testing, the amount of data is small and cannot reflect the situation of long-term monitoring. It is hoped that more test data can be added to better support the processing effect of temperature drift algorithm.

3. In the algorithm simulation experiment, the advantages of drift error compensation algorithm in this paper are verified by comparing the effect of multiple algorithms. However, the author only compares the advantages and disadvantages of the algorithm by standard deviation. So it is hoped that the author can make a comparison from more dimensions.

4. Figure 2, Figure 4, Figure 10 in this article are not clear enough, please upload the original image.

5. The author processed simulated signals under different conditions using different algorithms in Chapter 5.2. What is the conclusion and significance here? The author should conduct a detailed analysis in Chapter 5.2. In addition, I believe that the figures in this chapter cannot be well compared, such as what do the mean for every figure in Figure 5 ? The author should explain these figures clearly. Both Figure 6 and Figure 7 require careful analysis and explanation.

6. In the part of field experiment and analysis, the author's conclusion should be compared with other existing methods to illustrate the progressiveness of this method.

none

Reviewer 2 Report

The manuscript presents a drift-error compensation algorithm based on wavelet decomposition, which can adaptively separate the background and signal, and then calculate the frequency difference to compensate for the drift-error. The proposed seawater refractive index measurement is based on the laser heterodyne interferometer technique.

The topic is interesting and I believe this manuscript is worthy of publication in the Sensors. However, some issues should be addressed:

1. There are some uncertainties that affect the accuracy of the refractive index of seawater. The uncertainty in wavelength measurement and uncertainty in phase measurement can increase the error. This issue should be cleared and addressed.

2. The wavelength of the laser is considered as 632.8 nm. It should be mentioned if a stabilized He-Ne laser was used in the setup. Otherwise, I feel fluctuations in the power and frequency can reduce the accuracy.

3. The use of a super-heterodyne interferometer results in an increase in the resolution, and therefore, the refractive index measurement, by a factor of 2. It is recommended to mention the high-resolution structure in the literature. Two references are also suggested: 10.1088/0957-0233/16/9/017 and 10.1049/iet-opt:20060107

4. The Mach-Zehnder interferometry (MZI) is widely used for refractive index measurement. Please describe the advantages of the proposed setup and algorithm compared to the MZIs.

5. The manuscript needs to be carefully edited. For example line 24. References [5] and [6] should be cited in the context. Also, reference [10] is mentioned after [11].

6. Please give photodetector characteristics such as responsivity and spectral range.

Round 2

Reviewer 1 Report

The author made revisions to the article according to the suggestions, and I suggest to publish.

none

Reviewer 2 Report

The authors have tried to revise the manuscript according to previous comments. The revised version can be accepted for publication in Sensors, however, there are some minor revisions:

1. The unit of noise equivalent power (NEP) is watts per square-root-hertz, and therefore, the unit needs to be corrected in Table 1.

2. The caption of Table 1 must be corrected as "The characteristics of the photodetector" (instead of Standard deviation after algorithm processing").

3. It is recommended to define the abbreviations used in the schematic diagram of the seawater refractive index measurement system in Figure Caption 1. Also, a list of all abbreviations at the end of the manuscript can be useful for readers.
